# Prevalence and Antibiotic Resistance of *Escherichia coli* Isolated from Raw Cow’s Milk

**DOI:** 10.3390/microorganisms13010209

**Published:** 2025-01-19

**Authors:** Roxana Ionela Drugea, Mădălina Iulia Siteavu, Elena Pitoiu, Cristina Delcaru, Ecaterina Monica Sârbu, Carmen Postolache, Stelian Bărăităreanu

**Affiliations:** 1Faculty of Veterinary Medicine, University of Agronomic Sciences and Veterinary Medicine of Bucharest, 050097 Bucharest, Romania; roxana.drugea@fmvb.usamv.ro (R.I.D.); madalina.siteavu@fmvb.usamv.ro (M.I.S.); 2Synevovet Laboratory, Ilfov County, 077040 Chiajna, Romania; elena.pitoiu@synevo.com; 3Faculty of Biology, University of Bucharest, 050663 Bucharest, Romania; cristina.delcaru83@gmail.com (C.D.); ecaterina-monica.sarbu@drd.unibuc.ro (E.M.S.); carmen.postolache@bio.unibus.ro (C.P.)

**Keywords:** drug resistance, dairy farms, bovine mastitis, Romania, public health

## Abstract

*Escherichia coli (E. coli)* is one of the most common pathogens in both humans and livestock. This study aimed to investigate the prevalence of *E. coli* isolated from raw cow milk and evaluate its antimicrobial resistance rates. A total of 1696 milk samples were collected from Romanian dairy farms from 2018 to 2022. *E. coli* was isolated on various selective agar media, such as Cled agar and Columbia Agar with 5% Sheep Blood. The identification of *E. coli* was performed by MALDI-TOF MS. *E. coli* isolates were tested for their susceptibility against 18 commonly used antibiotics in a disk diffusion method. The overall prevalence of *E. coli* was 22.45% of all isolated pathogens. Antibiogram analysis revealed that 27.51% of *E. coli* isolates from milk were multidrug-resistant. Resistance was highest for penicillin–novobiocin (87.78%), followed by streptomycin (53.7%). Resistance to six drugs (amoxicillin, streptomycin, kanamycin–cephalexin, marbofloxacin, ampicillin) showed a significant increasing trend over time, while for two drugs (penicillin G-framycetin, doxycycline), a significant decrease was observed. Our results suggest that milk can be a reservoir of bacteria with the potential for infection in humans via the food chain. Furthermore, there is a need for surveillance and monitoring to control the increase in resistance to currently used antimicrobials in dairy farms because the occurrence of multidrug-resistant *E. coli* isolated from milk poses a health hazard to consumers.

## 1. Introduction

*E. coli* is a Gram-negative, non-spore-forming, flagellate, rod-shaped, and facultative anaerobic bacterium that ranges from being a normal resident of the gastrointestinal system in both humans and animals to being a significant pathogen [1,2]. *E. coli* is characterized as one of the most frequent causal agents associated with mastitis in dairy animals, with symptoms ranging from mild to severe clinical inflammation usually associated with toxemia, dysstasia, diarrhea, high fever, and decreased milk production [3,4,5,6]. Mastitis represents a serious problem in dairy cows worldwide, resulting in severe economic and financial losses for the cattle industry [7,8,9,10]. In addition, the bacteria responsible for mammary infections can be released into milk, along with their toxins, and thus transmitted to humans, causing a wide range of infections [9,11,12].

The appearance of pathogenic bacteria in milk leads to their multiplication due to milk being the perfect substrate for the growth of bacteria [13]. Although milk is highly nutritious and a staple food, the consumption of raw milk can affect public health, because it may contain numerous bacterial contaminants [1,14]. Several studies have reported the implications of milk and milk products in foodborne outbreaks caused by various pathogens, including *E. coli* [15,16]. *E. coli* is ubiquitous, and consequently, its presence in raw milk can come from sources such as dairy animals, the environment, or farm workers and milking equipment [17,18]. Furthermore, being harbored in the gastrointestinal tract of animals, *E coli* is well known as a hygienic indicator organism that reflects the fecal contamination of milk [19,20].

Based on its epidemiological, clinical, and pathogenic characteristics, *E. coli* can be classified into several different pathotypes, including enterotoxigenic *E. coli* (ETEC), enteropathogenic *E. coli* (EPEC), enteroinvasive *E. coli* (EIEC), enteroaggregative *E. coli* (EAEC), diffuse adherent *E. coli* (DAEC), Shiga toxin-producing *E. coli* (STEC), and enteroaggregative–hemorrhagic *E. coli* (EAHEC) [16,21,22]. Most *E. coli* strains are harmless; however, several *E. coli* strains have acquired specific virulence factors capable of causing several diseases in different sites [4]. For instance, *E. coli* strains that produce Shiga toxins are considered important foodborne zoonotic bacteria and the main risk foods for STEC infections in humans [11,23,24]. Even though STEC infections in ruminants are typically asymptomatic, these animals represent the primary reservoir of STEC and the consumption of unpasteurized milk has been associated with many foodborne outbreaks [18,24,25].

The overuse of antibiotics for prophylactic and curative purposes has led to the development of antimicrobial-resistant bacteria with negative effects on both public health and veterinary medicine [26,27,28,29,30]. According to the World Health Organization, one of the top ten threats to global human and animal health is antimicrobial resistance (AMR), and it is estimated that by 2050, it could be responsible for up to 10 million deaths annually and for increasing animal mortality rates to one percent per year in livestock [31].

In dairy farms, antimicrobial therapy is usually implemented to treat and prevent mastitis, and increasing antimicrobial resistance is attributed to the extensive and inappropriate use of antimicrobials to manage this disease [32,33]. As a consequence, in the last decade, several bacterial strains of *E. coli* isolated from dairy animals have gradually become resistant to different antimicrobial agents, and several publications have reported data concerning the increasing levels of acquired resistance to cefoxitin, sulphamethoxazole, cloxacillin, b-lactamase, tetracycline, quinolone, cephalosporin, amoxicillin, penicillin, cephalexin, trimethoprim, and chloramphenicol [34,35,36,37,38]. In addition, a significantly high degree of sensitivity rates to ciprofloxacin, gentamycin, oxytetracycline, levofloxacin, colistin, imipenem, enrofloxacin, gentamicin, and florfenicol has been recorded [35,39,40]. In various studies on AMR, *E. coli* is a common target due to its complex distribution in nature and the possibility of its transmission to humans, animals, food animals, plants, and wildlife [31].

Even though *E. coli* is frequently isolated from milk and dairy products, and despite the significant role the dairy industry plays in the development and spread of drug resistance, in Romania, there are limited reports on the antibiotic resistance profiles of this pathogen isolated from milk. Therefore, this study aims to investigate the prevalence and the resistance profiles of *E. coli* obtained from raw cow milk in Romania.

## 2. Materials and Methods

### 2.1. Milk Sampling

In the present study, bacteria were cultured from a total of 1696 raw cow’s milk samples from bovines with subclinical and clinical mastitis collected from 2018 to 2022. The samples were collected from 24 farms from various regions of Romania. These year-round farms housed herds ranging from 300 to 4500 cattle (including Holstein Friesians and Romanian Spotted Breed), of which 200 to 1600 were milking cows. The animals were housed in free-stall barns. This investigation was carried out in the Synevovet Laboratory in Bucharest, Romania. Raw cow milk samples were collected from the udders of cows. The milk was then transferred to a sterile container and kept refrigerated during transportation from the farms to the laboratory.

### 2.2. Microbial Isolation and Identification

For the detection of *E. coli*, the milk was incubated at 35–37 °C under aerobic conditions (MMM Group, Munich, Germany) for 24–48 h on Cled agar and Columbia Agar with 5% Sheep Blood. Colonies with typical *E. coli* morphologies were selected and identified using Matrix-Assisted Laser Desorption Ionization–Time-of-Flight Mass Spectrometry (MALDI-TOF MS, Brucker Daltonics, Bremen, Germany).

### 2.3. Antibiotic Sensitivity Test

The antimicrobial sensitivity profiles of *E. coli* were determined by the disk diffusion test applied on Muller–Hinton agar. Antimicrobial susceptibility testing (AST) disk diffusion plate readings and the interpretation of zones of inhibition as clinical categorizations were performed with the ADAGIO System (Bio-Rad, Marnes-la-Coquette, France, 3.1). A total of 18 antimicrobial agents (amoxicillin, ampicillin, amoxicillin–clavulanic acid, cefoperazone, cefquinome, ceftiofur, penicillin G-framycetin, penicillin–novobiocin, enrofloxacin, marbofloxacin, doxycycline, tetracycline, oxytetracycline, gentamicin, streptomycin, neomycin, kanamycin–cephalexin, and sulphamethoxazole–trimethoprim) were chosen based on their common usage for treatment on dairy farms. The interpretation of antibiotic resistance profiles was according to the guidelines of the Clinical and Laboratory Standards Institute (CLSI), and the isolates were categorized as susceptible, intermediate, or resistant to each antimicrobial agent, as shown in Table 1 [41,42].

### 2.4. Statistical Analysis and Determination of Multiple Antibiotic Resistance (MAR) Index

We entered all obtained data into Microsoft Excel 2013^®^ (Microsoft, Redmond, WA, USA) and computed all frequencies and percentages using GraphPad Prism software version 8.0 (GRAPH PAD Software Inc., San Diego, CA, USA). Statistical analyses for this study were conducted using linear regression (parametric approach) from SAS OnDemand for Academics with *p*-values less than 0.05 considered statistically significant. Figures were created with GraphPad Prism and Microsoft Office.

MAR index was calculated using Microsoft Excel in order to determine the level of antibiotic resistance of individual bacterial isolates using the formula reported by Manyi-Loh et al. [43] by dividing the number of antibiotics to which the isolate was resistant by the total number of antibiotics the isolate was exposed to in this study. A MAR value > 0.2 was indicative of multiple-antibiotic-resistant bacteria [43].

## 3. Results

### 3.1. Prevalence of E. coli in Milk Samples

During our 5-year study, 1696 raw milk samples from various Romanian dairy farms were analyzed, of which 980 (57.78%) were positive and 716 (42.22%) were negative. The detectable rate of bacterial pathogens from cow milk samples was 57.78%. We also found the co-occurrence of two species in 40 samples (4.08%), and for this reason, the total number of isolates was 1020. The results indicated the presence of *E. coli* in 229 (22.45%) of the isolated pathogens. Therefore, a total of 229 *E. coli* isolates were retained for additional antimicrobial resistance characterization.

### 3.2. Antibiotic Susceptibility of E. coli in Milk Samples

Our study demonstrated that *E. coli* isolates from raw milk in Romanian dairy farms had a variable degree of resistance to the tested antimicrobials. The results showed that 4.13%, 18.33%, and 77.54% of the *E. coli* isolates were intermediately resistant, resistant, and susceptible, respectively. Table 2 shows the average percentages of the antimicrobial susceptibility of *E. coli* isolates to specific antimicrobials between 2018 and 2022.

The resistance rates of all the investigated strains to the tested antibiotics ranged from 0 to 87.78%. The resistance of *E. coli* isolates to penicillin–novobiocin was the most serious, with a resistance rate of 87.77%, followed by resistance to streptomycin and ampicillin, with rates of 53.7% and 26.20%, respectively. The resistance rates of *E. coli* strains to other antimicrobials were between 2.18% and 22.71%, except for amoxicillin–clavulanic acid (0.44%). *E. coli* isolates were more susceptible to amoxicillin–clavulanic acid, ceftiofur, cefquinome, and quinolones such as marbofloxacin and enrofloxacin than to other tested antimicrobials, and the susceptibility rates were 99.13%, 97.38%, 93.90%, 93.89%, and 90.39%, respectively. A total of 77.54% of the isolates were sensitive to all antimicrobials tested, and only 4.13% of the isolates showed intermediate susceptibility.

The resistance rates and trends in the resistance of *E. coli* isolated from 2018 to 2022 to 18 individual antimicrobials are presented in Table 3.

Over our 5-year study period, the isolates showed an increasing trend of resistance to amoxicillin (from 3.77% to 19.05%), ampicillin (from 5.56% to 28.57%), streptomycin (from 18.52% to 78.38%, statistically significant with *p*-value < 0.05) (Figure 1), kanamycin–cephalexin (from 12.96% to 46.67%), marbofloxacin (from 0% to 14.29%), and gentamicin (from 1.85% to 9.76%). On the other hand, a decrease over time was observed for cefquinome (from 3.70% to 0%), and penicillin G-framycetin (from 5.56% to 3.92%, 2018–2021). Meanwhile, *E. coli* exhibited great resistance to penicillin–novobiocin throughout the investigation, with no significant increases in resistance seen. On the contrary, *E. coli* strains were highly susceptible to amoxicillin–clavulanic acid and ceftiofur, and there were no differences in sensitivity for these two antibiotics between the study periods.

A total of 27.51% of the *E. coli* isolates from Romanian dairy farms were found to be multidrug-resistant. Figure 2 shows that 70 strains were resistant to one antibiotic. Similarly, 61 strains were resistant to two antibiotics, followed by 22 and 14 strains resistant to three and four antibiotics, respectively. Furthermore, three bacterial strains were the most resistant, exhibiting resistance to 13 antibiotics, and nine and five strains were resistant to 12 and 11 antibiotics, respectively. This finding suggests that the widespread antibiotic resistance of *E. coli* strains recovered from raw milk should be regarded as a danger for dairy farms.

## 4. Discussion

The presence of pathogenic microorganisms in raw cow’s milk may be a risk to public health and accounts for approximately 90% of all dairy-related diseases [44]. The introduction of unwanted microorganisms into the dairy chain can occur from various sources, such as the environment, milking equipment, milk handlers, feed, and contaminated water or utensils [45,46,47]. In addition, some pathogens can contaminate raw cow’s milk intramammarily during mastitis (clinical or subclinical), which is considered the most common infectious disease encountered in dairy cattle, with serious consequences for the dairy industry worldwide [32]. Furthermore, mastitis has serious zoonotic potential and has been associated with the increasing development and rapid emergence of multidrug-resistant strains globally [36]. Many studies reported that bacteria are still the main causative pathogens of dairy cow mastitis and that *E. coli* is one of them [5,8,22]. *E. coli* is one of the most important opportunistic pathogens of humans and animals, responsible for a wide range of infections [46]. Accordingly, for milk hygiene and safety, it is important to ensure its microbial quality [35].

The increasing use of antibiotics in veterinary medicine contributes to the emergence of antibiotic resistance in microorganisms, which affects animal health and can have a negative impact on society [48]. Therefore, due to the bacterial diversity and the aggressive use of antibiotics with a broad spectrum of activity against both Gram-negative and Gram-positive organisms in dairy farms for therapeutic, metaphylactic, and prophylactic purposes, bacteria such as *E. coli* are becoming increasingly resistant to antibiotics [45]. In veterinary medicine, antibiotics such as tetracycline, ampicillin, and sulfamethoxazole-trimethoprim are widely used due to their low cost and availability for the treatment of infections [16].

To explore the potential risk associated with the consumption of raw milk in Romania, the occurrence and resistance of *E. coli* in raw milk samples from Romanian dairy farms were evaluated. Our findings showed that there was broad variation in the frequency and antibiotic susceptibilities of *E. coli* isolated from raw cow milk in Romania, and the results of this study should be beneficial for developing appropriate preventative and treatment methods in dairy farms.

In this study, 22.45% of the samples were found to be contaminated with *E. coli*, which agreed with several studies reporting *E. coli* contamination levels in raw milk of 11.1% [48], 12.1% [39], 26.0% [49], 48.9% [50], 30.16% [51], 33.8% [2], 34.4% [21], 68.0% [52], 78% [37], 70.4% [53], and 92.4% [36].

The prevalence of *E. coli* in raw milk found in this study is almost the same as that in a report from the Rajshahi Metropolitan area of Bangladesh. Sultana et al. [49] collected 50 raw milk samples from 10 dairy farms and found that 13 (26.0%) were *E. coli*-positive samples. They concluded that the prevalence of *E. coli* in raw milk is significant and may indicate several deficiencies related to poor hygiene during milking, irregular washing and sterilization of milking equipment, poor hygiene of milkers’ hands and animal udders, and infections of dairy cows. At the same time, these observations are in agreement with the results reported by Fard et al. [54], who showed that the contamination of raw milk with *E. coli* in developing countries is remarkable and identified 31 *E. coli* strains (20.6%) in 150 raw cow’s milk samples in Borujerd, Iran [54].

The highest prevalence of *E. coli* contamination of raw milk samples was identified in a study conducted in the Czech Republic by Skočková et al. [26], who isolated *E. coli* from 243 out of 263 milk samples (92.4%). It has been suggested that the presence of *E. coli* in raw cow’s milk is very common and it varies between places of production. Also, a high level of *E. coli* contamination was found in a study conducted by Hassani et al. [37], in which out of 100 milk samples randomly collected in the northwest of Iran, 78% were contaminated with *E. coli*, as well as in a study conducted by Tyasningsih et al. [53], in which 250 raw milk samples were collected from five dairy farms in East Java, Indonesia, and 176 (70.4%) were *E. coli*-positive samples.

On the other hand, the results of the present study are relatively higher as compared to a study by Yu et al. [48], in which 83 strains of *E. coli* were isolated from 750 raw milk samples collected in China in 2016, with a prevalence of 11.1%. They attributed the decrease in the prevalence of *E. coli* in raw milk samples to substantial structural changes in the Chinese dairy industry and the rapid growth of large herds. Another study reported that the prevalence of *E. coli* in raw milk collected from dairy cattle farms and collector’s bulk tanks in Sebeta town, Ethiopia, was 14/142 (9.9%) [55].

Several factors (e.g., sample size, farming system, farm size, milking equipment, milking method, location, ecology, duration of milk transit, and sanitary conditions) can be attributed to the variance in the prevalence of *E. coli* species in previous studies [55,56]. The prevalence of *E. coli* in our study was likely influenced by each farm’s unique characteristics, such as herd size (from 300 to 4500 cattle), farming system (free-stall barns), milking equipment and milking method, infrastructure, location and ecology, sanitary conditions, and duration of milk transit. Herd size and the potential for overcrowding in free-stall barns, as well as the sanitary conditions and the maintenance of the milking equipment, may have been significant contributors. Furthermore, environmental factors and the duration of milk transit might have played a role in the overall risk. Thus, the presence of *E. coli* in raw cow’s milk suggests potential public health hazards when milk is consumed without pasteurization, whether in rural or urban environments. Furthermore, besides contaminated milk, the spread of zoonotic bacterial diseases from animals to humans can also occur through the direct fecal–oral route, through improper food handling and cooking, or through people working with infected animals who can potentially contract zoonotic bacterial pathogens [12,57].

The development of antimicrobial-resistant bacteria is another important observation that poses a problem of high concern. The current study shows that, although the prevalence of the resistant *E. coli* strains isolated differed during our 5-year study, resistance to penicillin–novobiocin and streptomycin, as well as resistance to one or more antibiotics, was frequently detected.

*E. coli* isolates showed a high resistance to penicillin–novobiocin (87.77%), followed by resistance to streptomycin (53.7%) and ampicillin (26.20%). This result is in agreement with results reported by Momtaz et al. [58], who studied the antibiotic resistance pattern of *E. coli* in milk samples from bovines and found the highest resistance to penicillin (100%) [58]. Moreover, another study also reported a high level of streptomycin resistance (57.9%) in *E. coli* isolates from dairy products in Isfahan, Iran [16], and a study of dairy cattle in Jordan found that *E. coli* isolates exhibited high resistance to streptomycin (47.5%) and ampicillin (34.2%) [59]. Another similar resistance pattern of *E. coli* strains was found in raw bovine milk samples from the northwest of Iran, with 100% resistance to penicillin and 30.1% to ampicillin [37]. In India, *E. coli* collected from milk samples appeared to be as resistant to amoxicillin–clavulanic acid (83.33%) and gentamicin (58.33%) as in our study [39]. In the western part of Romania, Pascu et al. [40] determined the bacteria responsible for bovine mastitis and analyzed their antimicrobial resistance. They found that *E. coli* represented 13.79% (16/116) of all isolated strains, and *E. coli* resistance to ampicillin, streptomycin, tetracycline, and sulfamethoxazole was observed most often. This demonstrates that our results regarding the prevalence and antibiotic resistance of *E. coli* in milk samples are similar to those of their study [40].

The present study showed that *E. coli* isolates were highly sensitive to amoxicillin–clavulanic acid (99.13%), ceftiofur (97.38%) cefquinome (93.90%), quinolones such as marbofloxacin (93.89%) and enrofloxacin (90.39%), and gentamycin (89.52%). Similarly to the present study, a high sensitivity to ceftiofur and gentamycin was reported [6,7,17,21,52,56].

Regarding the antimicrobial resistance rates and the resistance trends of *E. coli* isolated from milk samples in Romania from 2018 to 2023, in our study, we identified an increasing trend of resistance to amoxicillin (from 3.77% to 19.05%), ampicillin (from 5.56% to 28.57%), streptomycin (from 18.52% to 78.38%), kanamycin–cephalexin (from 12.96% to 46.67%), marbofloxacin (from 0% to 14.29%), and gentamicin (from 1.85% to 9.76%). Cefquinome and penicillin G-framycetin resistance decreased over time from 3.70% to 0% and from 5.56% to 3.92%, respectively. Amoxicillin and ampicillin saw significant increases in resistance. This could be related to an increased or inappropriate use of these antibiotics, which are commonly used in both human and veterinary medicine, in livestock. An overuse and misuse of these antibiotics, particularly in intensive farming systems, can promote the emergence and selection of resistant bacterial strains. Resistance to streptomycin and kanamycin–cephalexin also showed considerable increases. These antibiotics are frequently used in veterinary practice to treat infections in cattle, which might lead to selective pressure for resistant strains. Marbofloxacin and gentamicin resistance increased over time as well. These quinolones and aminoglycosides are critical antibiotics used to treat severe infections in both animals and humans, and increasing resistance to these agents is concerning due to their importance in human medicine. On the other hand, cefquinome and penicillin G-framycetin showed a decreasing trend in resistance. This decrease in resistance could indicate a reduction in the use of these antibiotics or better adherence to regulations on antibiotic use in veterinary medicine. Furthermore, farmers and veterinarians may have become more aware of the risks of overusing certain antibiotics, leading to a more targeted use of these drugs. Not least, better hygiene and sanitation practices on farms might have reduced the bacterial load and minimized the selective pressure for resistance.

The increase in multidrug-resistant *E. coli* strains has become a serious public health problem worldwide. In this study, 63 out of 229 (27.51%) *E. coli* isolates were multidrug-resistant. Similar results were obtained in a study from Isfahan, Iran, in which an MDR profile was reported for 16 of 38 (73.3%) isolates collected from milk samples. The study suggested that the observation of MDR in *E. coli* isolates can be attributed to an irrational use of antimicrobial agents or genetic mutations [16].

Our study indicates that the presence of multidrug-resistant bacteria and the uncontrolled use of antibiotics in dairy farms have a negative impact on milk quality. In addition, antibiotic-resistant pathogens can also be transferred to humans through milk consumption and, consequently, complicate the treatment of infections. Therefore, choosing the appropriate antibiotic against *E. coli* strains is the most effective measure for the proper treatment of diseases in dairy farms.

## 5. Conclusions

In conclusion, our study enriches the knowledge regarding the prevalence and antimicrobial resistance of *E. coli* isolated from raw cow’s milk samples from different dairy farms in Romania. The results show that the isolation rate of *E. coli* strains from raw cow’s milk samples was 22.45%, which highlights the risk for *E. coli* infections in humans via the food chain associated with the consumption of unpasteurized milk. Furthermore, our findings suggest that resistance to multiple antimicrobial agents, with 27.51% of *E. coli* isolates being multidrug-resistant, is a major public health problem and requires solutions to reduce this phenomenon. The resistance of *E. coli* was highest against the combination of penicillin and novobiocin (87.78%). Possible explanations for this observation could include the widespread use of these antibiotics in both human and veterinary medicine, leading to the selection of resistant strains. Consequently, as *E. coli* impacts both animal health and human welfare, there is a need for rigorous surveillance of the bovine and human *E. coli* population, as well as for a prudent use of antimicrobials, extensive dairy farm hygiene, and biosecurity measures to avoid environmental pathogens and ascertain the security of the food chain, as well as for managing human and animal health.

## Figures and Tables

**Figure 1 microorganisms-13-00209-f001:**
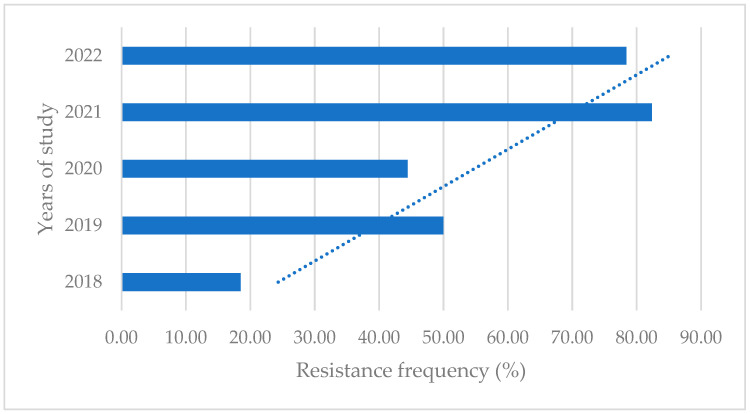
Rate of resistance to streptomycin. The trend of increasing resistance to this antibiotic is noticeable.

**Figure 2 microorganisms-13-00209-f002:**
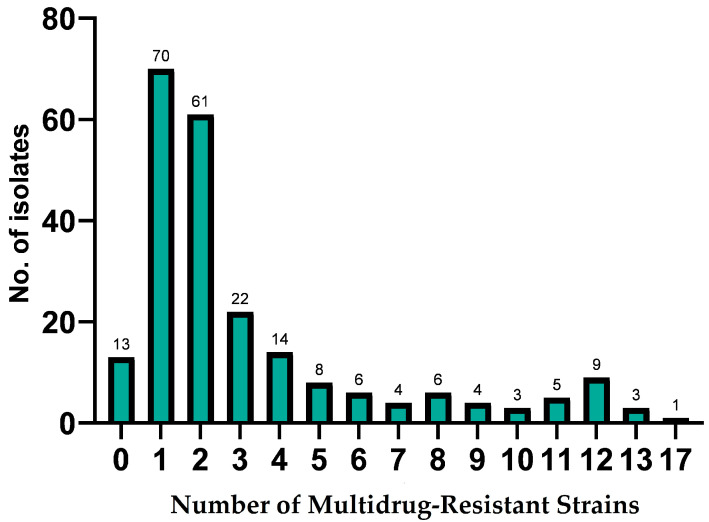
Multidrug resistance of 229 *E. coli* strains to 18 antibiotics. All 229 *E. coli* isolates were analyzed for multidrug resistance to amoxicillin, ampicillin, amoxicillin–clavulanic acid, cefoperazone, cefquinome, ceftiofur, penicillin G-framycetin, penicillin–novobiocin, enrofloxacin, marbofloxacin, doxycycline, tetracycline, oxytetracycline, gentamicin, streptomycin, neomycin, kanamycin–cephalexin, and sulphamethoxazole–trimethoprim.

**Table 1 microorganisms-13-00209-t001:** Criteria for the interpretation of antibiotic resistance profiles of *E. coli* isolates.

Antibiotics	Disk Content (µg)	Interpretive Categories and Zone Diameter Breakpoints (Nearest Whole mm)		
		Susceptible	Intermediate	Resistant
**β-lactamase**				
Amoxicillin (AMX)	10	≥17	14–16	≤13
Ampicillin (AM)	10	≥17	14–16	≤13
Amoxicillin–clavulanic acid (AMC)	20/10	≥18	14–17	≤13
Cefoperazone (CFP)	75	≥18	16–17	≤15
Cefquinome (CEF)	30	≥22	-	≤19
Ceftiofur (EFT)	30	≥21	18–20	≤17
Penicillin G-framycetin (PFY) *	10 UI **/100	≥18	16–17	≤15
Penicillin–novobiocin (PNV)	30	≥18	15–17	≤14
**Quinolones**				
Enrofloxacin (ENR)	5	≥23	17–22	≤16
Marbofloxacin (MAR)	5	≥20	15–19	≤14
**Tetracyclines**				
Doxycycline (DO)	30	≥15	12–14	≤11
Tetracycline (TE)	30	≥15	12–14	≤11
Oxytetracycline (OT)	30	≥15	12–14	≤11
**Aminoglycosides**				
Gentamicin (CN)	10	≥16	13–15	≤12
Streptomycin (S)	10 UI	≥18	16–17	≤15
Neomycin (N)	30	≥17	-	≤15
Kanamycin–cephalexin (KCL)	30/15	≥20	18–19	≤17
**Sulphonamides**				
Sulphamethoxazole–trimethoprim (Sxt)	23.75/1.25	≥16	11–15	≤10

Key: * The interpretation of antibiotic resistance profiles was according to the recommendations of experts at pharmaceutical companies; ** UI = international unit.

**Table 2 microorganisms-13-00209-t002:** Antimicrobial susceptibility of *E. coli* isolates from raw milk (*n* = 229) across 2018 to 2022.

Antibiotics	Resistance		Intermediate		Susceptible	
	#	%	#	%	#	%
**β-lactamase**						
Amoxicillin (AMX)	26	11.35	2	0.87	201	87.77
Ampicillin (AM)	60	26.20	7	3.06	162	70.74
Amoxicillin–clavulanic acid (AMC)	1	0.44	1	0.44	227	99.13
Cefoperazone (CFP)	7	3.47	21	10.40	174	86.14
Cefquinome (CEF)	7	3.29	6	2.82	213	93.90
Ceftiofur (EFT)	5	2.18	1	0.44	223	97.38
Penicillin G-framycetin (PFY)	15	7.58	21	10.61	162	81.82
Penicillin–novobiocin (PNV)	201	87.78	8	3.49	20	8.73
**Quinolones**						
Enrofloxacin (ENR)	16	6.99	6	2.62	207	90.39
Marbofloxacin (MAR)	14	6.11	0	0.00	215	93.89
**Tetracyclines**						
Doxycycline (DO)	40	17.54	10	4.39	178	78.07
Tetracycline (TE)	52	22.71	0	0.00	177	77.29
Oxytetracycline (OT)	51	22.27	1	0.44	177	77.29
**Aminoglycosides**						
Gentamicin (CN)	19	8.30	5	2.18	205	89.52
Streptomycin (S)	120	53.57	20	8.93	84	37.50
Neomycin (N)	44	19.21	11	4.80	174	75.98
Kanamycin–cephalexin (KCL)	46	21.20	39	17.97	132	60.83
**Sulphonamides**						
Sulphamethoxazole–trimethoprim (Sxt)	22	9.37	2	0.88	202	89.38
**Overall**	746	18.33	161	4.13	3120	77.54

Key: # = number of isolates; % = resistance frequency.

**Table 3 microorganisms-13-00209-t003:** Antibiotic resistance profile of *E. coli* isolates from raw milk (*n* = 229).

Antimicrobials	% of Resistant Isolates (No. of Tested Isolates)					
	2018	2019	2020	2021	2022	*p*-Value
**β-lactamase**						
Amoxicillin (AMX)	3.70 (54)	15.22 (46)	8.33 (36)	11.76 (51)	19.05 (42)	0.167
Ampicillin (AM)	5.56 (54)	28.26 (46)	13.89 (36)	52.94 (51)	28.57 (42)	0.264
Amoxicillin–clavulanic acid (AMC)	0 (54)	2.17 (46)	0 (36)	0 (51)	0 (42)	0.559
Cefoperazone (CFP)	3.70 (54)	4.35 (46)	2.78 (36)	3.92 (51)	6.67 (15)	0.179
Cefquinome (CEF)	3.70 (54)	4.44 (45)	2.78 (36)	3.92 (51)	0 (27)	0.179
Ceftiofur (EFT)	1.85 (54)	2.17 (46)	2.78 (36)	0 (51)	4.76 (42)	0.580
Penicillin G-framycetin (PFY)	5.56 (54)	13.33 (45)	5.56 (36)	3.92 (51)	16.67 (12)	0.552
Penicillin–novobiocin (PNV)	79.63 (54)	78.26 (46)	100.00 (36)	94.12 (51)	90.48 (42)	0.251
**Quinolones**						
Enrofloxacin (ENR)	0 (54)	13.04 (46)	5.56 (36)	3.92 (51)	14.29 (42)	0.387
Marbofloxacin (MAR)	0 (54)	10.87 (46)	5.56 (36)	1.96 (51)	14.29 (42)	0.370
**Tetracyclines**						
Doxycycline (DO)	9.26 (54)	32.61 (46)	8.33 (36)	23.53 (51)	12.20 (41)	0.938
Tetracycline (TE)	12.96 (54)	32.61 (46)	11.11 (36)	33.33 (51)	21.43 (42)	0.665
Oxytetracycline (OT)	12.96 (54)	28.26 (46)	8.33 (36)	35.29 (51)	24.39 (41)	0.473
**Aminoglycosides**						
Gentamicin (CM)	1.85 (54)	10.87 (46)	3.13 (36)	15.69 (51)	9.76 (41)	0.317
Streptomycin (S)	18.52 (54)	50.00 (46)	44.44 (36)	82.35 (51)	78.38 (37)	0.029 *
Neomycin (N)	11.11 (54)	23.91 (46)	11.11 (36)	29.41 (51)	17.95 (39)	0.530
Kanamycin–cephalexin (KCL)	12.96 (54)	21.74 (46)	13.89 (36)	19.61 (51)	46.67 (30)	0.144
**Sulphonamides**						
Sulphamethoxazole–trimethoprim (SxT)	3.79 (54)	19.57 (46)	0 (36)	11.76 (51)	12.82 (39)	0.731

*—statistically significant differences.

## Data Availability

The original contributions presented in this study are included in the article. Further inquiries can be directed to the corresponding author.

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
