# Peer review of "Prevalence and Antibiotic Resistance of Escherichia coli Isolated from Raw Cow’s Milk"

_microorganisms, 2025, doi:10.3390/microorganisms13010209_

Round 1
Reviewer 1 Report
Comments and Suggestions for Authors
General comments:
The introduction is well described and the problem is clear. The methodology needs some adjustments, mainly a tendency test for resistance increase/decrease over the years and an inclusion of herds description. Another point, once you are evaluating drug resistance, which are the most common antibiotics used in a Romanian scenario or even in those sampled herds? The results are, basically, descriptive but adequate. Discussion should be improved by trying to present some explanation for your results and not being only a results comparison to other studies in general.
Specific comments:
Line 14: "22.45% of all".. not "in all"
Line 23: All your keywords are included in your title. You need to choose other ones instead.
Line 33: correct to "industry [7–10]"
Line 52: correct to "sites [23]"
Lines 76-77: Join your objective with the paragraph.
Lines 81-82: How many farms and their characteristics? Breed? Number of cows? Rearing systems?
Line 102: Table 1 is not mentioned in the text. What is "I", "R", and "S" in that table?
Line 109: Which statistics test you are using that involves a p-value? None.
Line 126 and 127: I think that you do not need those Tables 2 and 3. You can only show such information in your text.
Figures 1 to 6: You need to standardize it showing or not showing the label for all of them and the same legend format.
Lines 151-156: Are you sure about those %? In any graph the "x" axis does not reach>70%, so, how could you indicate in that graph a maximum of 78.38% of resistance for streptomycin, for example? Also, you need to indicate a legend for the "x" and "y" axes. Another point, you should present a p-value for your "trend test". You can calculate it from a linear regression (parametric approach) or a non-parametric test, such as the Jonckheere-Terpstra test. I will provide a SAS script with a non-parametric test attached for the data on Amoxicillin resistance (the p-value resulted in 0.05, i.e. significative for a tendency of increase as you mentioned). Do it for all the antibiotics tests and present in figures just the significative ones.
Line 186: Correct to "[38]. Many"
Line 190: Correct to "quality [36]."
Lines 211, 217, 221, 224, 226, 230 ... : Citations not formatted based on the journal's guidelines. You can cite like this "Sultana et al. [paper number]" or "Sultana and collaborators [paper number]".
Line 218: Correct to: "E. coli"
Lines 236-237: You should include some characteristics of the herds you evaluated to support your explanation of which factors might contribute to the spread of E. coli in your case.
Line 261: You need to adequately address citations like that in the test.
Lines 271-277: You need to explore why those effects were found.
Line 283: Correct to "mutations [52]."

Reviewer 2 Report
Comments and Suggestions for Authors
Dear editor, dear authors,
I have read carefully the paper entitled “Prevalence and antibiotic resistance of Escherichia coli isolated from raw cow’s milk”. The paper covers the important topic of growing antibiotic resistance and prevalence of E. coli in samples of raw cow’s milk, that could lead to serious problems in human and animal health and mentions some possible solutions to resolving this issue in the future.
However, there are some issues throughout the text that need revision. My suggestion is that the paper should be accepted after minor revisions.
SPECIFIC COMMENTS
1. Line 47: Authors mentioned five different pathotypes of E. coli, but seven of them were listed. It should be changed to “…several different pathotypes…”.
2. Line 50: E. coli is not in italic here, it should be corrected.
3. In Materials and Methods sections 2.2., 2.3. and 2.4. the location (name, city, country) of companies that produced agars and instruments should be added in parentheses. Authors should check and add where needed.
4. Line 93: Since AST is first mentioned here, the authors should indicate its full name, and the abbreviation in parentheses.
5. Line 100: Same as point 4., the authors should indicate the full name of CLSI and add the abbreviation in parentheses.
6. Authors should refer to Table 1 in the text before it appears.
7. Table 1: Authors should add the meaning of UI in Table footnote.
8. Table 4: Authors should add the meaning of # in Table footnote.
9. Figures 1-6: Authors should add the meaning of I, S and R in Figure descriptions.
10. Line 210: Word “almost” is not needed, because “similar” already implies a degree of resemblance. Or it could be corrected to “almost the same” for example.
11. Line 253: “in milk samples from bovines’ samples” should be changed to “in milk samples from bovines” for more clarity.
12. Lines 276 and 277: “Respectively” should be moved to the end of this sentence. “Cefquinome and penicillin G-framycetin have decreased over time from 3.70% to 0% and from 5.56% to 3.92%, respectively”.
13. References 16 and 52, and 51 and 53 are the same, authors should check all the references and correct if there are more same ones.

Reviewer 3 Report
Comments and Suggestions for Authors
The study aims to investigate trends in antibiotic resistance development among different strains of Escherichia coli isolated from raw cow's milk. I have a few points that I would like to bring to the attention of the authors.
1. Please provide information on the number of farms that participated in the study.
2. Please provide more details regarding the test statistics used to calculate p-values in your research.
3. Table 4. Which year does this data represent, or is it an average for all the years between 2018 and 2022?
4. Lines 151-160. Please indicate which of the listed differences in the percentage of isolated antibiotic-resistant strains were statistically significant.
5. Figure 7. Please specify the designations for the numbers on the x-axis in the figure legend.
6. How can the authors account for the fact that the highest levels of resistance to penicillin-novobiocin have been identified? Could this be a result of the increased use of these antibiotics in animal farming?
7. Lines 279-280: “In this study, 63 out of 220 (27.51%) E. coli isolates were multi-drug resistant.”
I did not find any information regarding the 63 strains mentioned in the article, such as what antibiotics they were resistant to and how many antibiotics were not effective against each strain. There is a discrepancy in the reported total number of strains. Tables 4 and 5, as well as Figure 7, indicate a total of 229, while here the reported value is 220.
8. The conclusions do not indicate a high prevalence of resistant strains to the combination of penicillin and novobiocin. It would be beneficial to notice possible explanations for this observation and measures that could be taken to address it.
